# Clinical Characteristics, Outcomes, and Risk Factors for Mortality in Patients with *Stenotrophomonas maltophilia* Bacteremia

**DOI:** 10.3390/jcm11113085

**Published:** 2022-05-30

**Authors:** Siripen Kanchanasuwan, Jakkapan Rongmuang, Pisud Siripaitoon, Narongdet Kositpantawong, Boonsri Charoenmak, Thanaporn Hortiwakul, Ozioma Forstinus Nwabor, Sarunyou Chusri

**Affiliations:** 1Department of Internal Medicine, Faculty of Medicine, Prince of Songkla University, Songkhla 90110, Thailand; kaymed29@yahoo.com (S.K.); jakky_frog@hotmail.com (J.R.); grippen45@gmail.com (P.S.); poom_032@yahoo.com (N.K.); cboonsri@medicine.psu.ac.th (B.C.); hratri@medicine.psu.ac.th (T.H.); nwaborozed@gmail.com (O.F.N.); 2Natural Product Research Center of Excellence, Faculty of Science, Prince of Songkla University, Hat Yai, Songkhla 90110, Thailand; 3Department of Biomedical Sciences, Faculty of Medicine, Prince of Songkla University, Songkhla 90110, Thailand

**Keywords:** *Stenotrophomonas maltophilia*, characteristics, outcomes, mortality, bacteremia

## Abstract

This study aimed to establish the clinical features, outcomes, and factors associated with mortality in patients with *Stenotrophomonas maltophilia* (*S*. *maltophilia*) septicemia. The characteristics and outcome data used in this retrospective study were collected from medical records at Songklanagarind Hospital. Risk factors for survival were analyzed using *χ*^2^-tests, Kaplan–Meier curves, and Cox regression. A total of 117 patients with *S**. maltophilia* bacteremia were analyzed. The patients’ median age was 45 years, 77 (70%) were male, 105 (90%) had comorbidities, 112 (96%) had previously undergone carbapenem therapy, and over half of the patients were on invasive medical devices. Trimethoprim-sulfamethoxazole (TMP-SMX) and fluoroquinolone showed high susceptibility rates to *S*. *maltophilia*, with 93% and 88% susceptibility, respectively. Patients who received appropriate empirical antibiotic treatment had significantly reduced 14-day, 30-day, and in-hospital mortality rates than those who did not (*p* < 0.001). The days of hospital stay and costs for those who received appropriate and inappropriate empirical antimicrobial treatment were 21 and 34 days (*p* < 0.001) and 142,463 and 185,663 baht, respectively (*p* < 0.002). Our results suggest that an appropriate empirical antibiotic(s) is significantly associated with lower 30-day mortality in hospitalized patients with *S*. *maltophilia* septicemia.

## 1. Introduction

*Stenotrophomonas maltophilia* is a motile, glucose, non-fermenting, gram-negative bacterium that widely exists in hospital environments [1,2]. In the last few decades, it has been recognized as a significant pathogen responsible for hospital acquired infections in severely debilitated or immunocompromised individuals [3,4,5,6,7,8]. It usually causes pneumonia and bacteremia and, less frequently, it can cause infections of the urinary tract, intra-abdominal organs, and wounds [2,9].

Due to its inherent resistance to several antibiotic classes, treating *S**. maltophilia* infections is difficult. *S**. maltophilia* employs a wide spectrum of resistance mechanisms, including reduced permeability, expression of multidrug efflux pumps, and the synthesis of beta-lactamase, carbapenemase, and aminoglycoside-modifying enzymes [10,11]. Moreover, distinguishing between infection and colonization is challenging because of the bacterium’s ability to colonize respiratory epithelial cells and invasive medical device surfaces. Hence, the administration of appropriate antibiotics is occasionally delayed, leading to high mortality [10,11,12]. TMP-SMX is the preferred antibiotic for treating *S**. maltophilia* infections [13,14]. However, fluoroquinolones and minocycline are also alternative management options [15,16,17]. According to a systematic review, infections with *S**. maltophilia* are associated with high rates of mortality, ranging from 21 to 69% [18]. Prior reports have demonstrated the risk factors associated with mortality among *S**. maltophilia* bacteremia patients, including septic shock, an Acute Physiology and Chronic Health Evaluation (APACHE) II score more than 15 points, indwelling invasive medical devices, mechanical ventilation, admission to intensive care units (ICU), prolonged hospitalization, and concurrent administration of immunosuppressive agents or broad-spectrum antibiotics [19,20].

To date, information regarding the clinical characteristics and outcomes of patients with *S**. maltophilia* bacteremia in Thailand has rarely been reported. Thus, this study reviewed the data of patients with *S**. maltophilia* septicemia at Songklanagarind Hospital to establish the clinical characteristics, outcomes, and risk factors associated with mortality. We find that treatment with appropriate empirical antimicrobial agent(s) is significantly associated with a lower 30-day mortality rate, and that a high APACHE II score is a significant independent predictor of mortality, suggesting that *S**. maltophilia* bacteremia should always be considered in patients who exhibit potential risk factors.

## 2. Patients and Methods

### 2.1. Patients and Setting

This study was conducted retrospectively, using data collected at a tertiary care hospital in southern Thailand from 2010 to 2020. All hospitalized patients aged >15 years who had one or more episodes of infection with *S*. *maltophilia* isolated from blood culture were eligible for this study. Patients’ clinical information was obtained from medical records, while microbiological data were extracted from a laboratory database. Only *S*. *maltophilia* septicemia as a first episode was included in the analysis to eliminate case duplication.

### 2.2. Bacterial Identification and Antimicrobial Susceptibility

Conventional biochemical tests, according to Bergey’s Manual of Systematic Bacteriology, were performed to identify *S*. *maltophilia*. Matrix-Assisted Laser Desorption/Ionization-Time of Flight (MALDI-TOF) mass spectrometry (MS) was further employed to confirm the identifications. The Kirby-Bauer disk diffusion method was used for antimicrobial evaluation of TMP-SMX and levofloxacin, while the Broth microdilution method was performed when testing ceftazidime and chloramphenicol. The results were interpreted according to the Clinical and Laboratory Standards Institute (CLSI) [21,22]. Regarding polymyxin susceptibility testing, there are no currently established polymyxin breakpoints for *S*. *maltophilia*, but Minimal Inhibitory Concentration (MIC) of ≥4 µg/mL is often considered resistance [23].

### 2.3. Study Design and Data Collection

#### Data Retrieval

Patients’ electronic medical records were reviewed. Demographic and clinical variables including age, sex, body mass index, comorbidities, and immunocompromised status were obtained. Comorbidities included cerebrovascular disease, cardiovascular disease, chronic kidney disease, and diabetes mellitus. Immunocompromised status was defined as those who had an absolute neutrophil count < 0.5 × 10^9^/L for >two weeks or were on immunosuppressive therapy (chemotherapy within six weeks or corticosteroids at a dosage greater than 15 mg of prednisolone daily for >two weeks or disease-modifying antirheumatic drugs within four weeks after bacteremia). Clinical information included instances of pneumonia, initial ICU admission, the patient’s APACHE II score, their history of previous exposure to antibiotics, and the use of invasive medical devices. Microbiological data included MICs. Treatment data included the appropriateness of empirical antimicrobial therapy and the duration of antibiotic treatment. An appropriate empirical antibiotic was defined as the prescription of drug(s) that are effective against *S*. *maltophilia* isolates within 72 h of blood culture collection. TMP-SMX and fluoroquinolones are considered treatment options for infections with *S*. *maltophilia* [24]. The primary outcome was 30-day mortality, whereas secondary outcomes were in-hospital mortality, 14-day mortality, length of hospital stay, and hospital expenses. The length of in-hospital stay was defined as the duration of hospital stay after *S*. *maltophilia* septicemia was diagnosed. Hospital expenses were divided into two categories: antimicrobial pharmaceutical costs and hospitalization-associated costs.

### 2.4. Statistical Analysis

Data on demographics and clinical outcomes were analyzed. Continuous variables were reported as means ± SD (standard deviation), and categorical variables were reported as percentages (%) or frequency counts (n). Continuous variables were tested using the D’Agostino-Pearson test to check for normality, and if data were normally distributed, Student’s *t*-tests were conducted and data were presented as means ± SD. Otherwise, the Mann–Whitney test was performed, and data were presented as medians (IQR). For categorical variables, *χ*^2^-tests or Fisher’s exact test were used. Univariate analysis was used to determine the crude odds ratio (OR) for mortality. A multivariate logistic regression model was used for 30-day mortality-related variables and variables with *p*-values ≤ 0.2. The significance level was set at *p* ≤ 0.05. The adjusted OR and 95 percent confidence interval were used to express the relationship between variables and outcomes. A survival analysis using Cox proportional hazard regression was done to determine the factors associated with survival.

### 2.5. Ethical Statement

The institutional review board of the Faculty of Medicine, Prince of Songkla University, Thailand, approved this retrospective study (REC: 58368144). The authors were given permission to retrieve clinical and microbiological data from the hospital database with a consent waiver. Before being assessed and used, all data were anonymized. The researchers confirm that we conducted this research in line with the Declaration of Helsinki principles.

## 3. Results

From 2010 to 2020, 129 blood samples from 129 patients yielded positive cultures for *S**. maltophilia*. Two or more organisms were isolated from one blood culture of 12 (9.3%) patients; therefore, they were excluded from the study. Figure 1 depicts the patient enrollment in this study. Of 117 patients with *S*. *maltophilia* septicemia, 54 (46.2%) received appropriate empirical antibiotic(s).

The records of 117 *S*. *maltophilia* bacteremia patients were included in the analysis. Table 1 reveals the baseline demographic characteristics and comparison of features between patients receiving and not receiving appropriate empirical antibiotics.

The patients’ median age was 45 years (range, 40–51 years), and 66% (77/117) were men. *S*. *maltophilia* was predominantly found in hospitalized patients, with comorbidities in 90% of patients with underlying diseases. However, only six (5%) patients were immunocompromised. Most patients (70%) had a body mass index ≥ 30 kg/m^2^, reflecting a high incidence of obesity among patients with *S*. *maltophilia* bacteremia. Also, 54 patients (46%) were critically ill, with an average APACHE II score of 19. In addition, over half (55%) of the patients had pneumonia and bacteremia. Most patients (96%) had a recent history of carbapenem exposure before *S*. *maltophilia* bacteremia, followed by cephalosporin, fluoroquinolone, beta-lactam/beta-lactamase inhibitor, and aminoglycoside exposure, with rates of 73%, 62%, 59%, and 34%, respectively.

We found significantly higher APACHE II scores, intensive care unit admission rates, rates of intravascular device and urinary catheter, and rates of empirical antimicrobial therapy with TMP-SMX, fluoroquinolone, and colistin for *S*. *maltophilia* bacteremia in patients who received appropriate antibiotic(s) than in those who did not. Other baseline characteristics, including demographic data, did not significantly differ between the two groups.

Only 54 (46.2%) patients received appropriate empirical antimicrobial agents, whereas 63 (53.8%) received inappropriate empirical antimicrobial agents. The majority of patients (94%) received empirical antibiotic therapy with carbapenem, which is ineffective against *S*. *maltophilia*, whereas one-third (34%) received empirical antibiotic treatment with colistin, and only one-fourth had received TMP-SMX or fluoroquinolone for empirical therapy. The duration of antibiotic treatment did not differ between the two groups.

Antimicrobial susceptibility testing of the *S*. *maltophilia* isolated from the 117 patients is shown in Table 2**.** The results indicated that TMP-SMX, levofloxacin, and chloramphenicol showed the highest susceptibility rate against *S*. *maltophilia*, with susceptibility values of 93%, 88%, and 82% respectively, compared with the 65% and 35% non-susceptible rate of ceftazidime and colistin. Thus, TMP-SMX was adopted as the most effective antibiotic against *S*. *maltophilia* bacteremia, followed by fluoroquinolone.

The clinical outcomes of the patients treated for *S*. *maltophilia* bacteremia are shown in Table 3**.** The 14-day, 30-day, and in-hospital mortality rates of those who received appropriate empirical antimicrobial treatment were significantly lower than the rates of those who received inappropriate treatment, with values of 7%, 7%, and 9% and 33%, 48%, and 51%, respectively (*p* < 0.001). Antimicrobial and non-antimicrobial expenses were the highest for patients who received appropriate antibiotic(s). The length of hospital stay from *S**. maltophilia* bacteremia onset was significantly lower for patients who received appropriate empirical antibiotic (s) at 21 days, compared to 34 days for patients who did not (*p* < 0.001). Moreover, hospital costs were significantly lower for patients who received appropriate empirical antibiotic(s) (142,463 and 185,663 baht, respectively; *p* = 0.002).

The factors associated with 30-day mortality in patients with *S**. maltophilia* bacteremia are shown in Table 4. A high APACHE II score was significantly associated with 30-day mortality (OR, 1.19; *p* = 0.012). The appropriate empirical antibiotic reduced 30-day mortality significantly (OR, 0.04; *p* = 0.001).

The 30-day Kaplan–Meier survival rates among patients with *S*. *maltophilia* who received appropriate empirical antibiotic(s) differed significantly from those of patients who did not (*p* < 0.001, log-rank test), as illustrated in Figure 2.

## 4. Discussion

Since the extensive prescription of antimicrobial agents in the past decade, *S**. maltophilia* has become the third most prevalent non-fermentative gram-negative bacterial isolate causing hospital acquired infections, preceded by *Pseudomonas aeruginosa* and *Acinetobacter* spp. *S**. maltophilia* infections. These are associated with high mortality rates ranging from 21 to 69%, and the overall 30-day mortality rate of 29% observed in this study is comparable to previous reports.

Several characteristics of *S**. maltophilia* bacteremia patients were similar to previous reports. Several reports showed that infection due to *S**. maltophilia* commonly occurred among the critically ill patients who were admitted in ICUs, while less than half (46%) of the patients in this research were primarily admitted in ICUs [3,4,5,6,7,8,12]. This can be explained with the constrained ICU capacity in our setting. The patients in this study had relatively severe illness with a median APACHE score of 19, and most were subjected to a mechanical ventilator (82%), intravascular devices (62%) and urinary catheterization (86%), respectively. Thus, the acquisition of this organism frequently occurred among mechanically ventilated patients [12]. Most of the patients in this study (90%) had several medical morbidities. Similarly, risk factors of this infection including multiple debilitating illnesses were described in a previous report. Additionally, previous exposure to several broad-spectrum antibiotics were previously demonstrated to be a risk factor for infection due to *S**. maltophilia*. This study shows that *S**. maltophilia* mostly affects patients with prior use of antibiotics, especially carbapenem (96%). This suggests that the use of carbapenem in patients who are hospitalized may lead to the emergence of nosocomial infections caused by *S**. maltophilia* as its intrinsic resistance to carbapenem and the selective growth of non-susceptible isolates. An antimicrobial susceptibility test in this study showed unfavorable resistance to commonly-used antimicrobial agents for nosocomial infection including ceftazidime and colistin. These findings are similar to previously published data [10,11]. The resistance seemed relatively low among levofloxacin (12%) and TMP-SMX (7%). The appropriate choices in our setting included TMP-SMX and fluoroquinolones. It is interesting that the resistance of colistin is relatively high at 35%. This finding is in line with the reports in several regions of the rising of colistin-resistant *S*. *maltophilia* [10,11,14]. Unlike the in vitro study of Enterobacteriaceae, the data on combination therapy of *S**. maltophilia* remains unclear [25].

Previous reports have demonstrated that risk factors associated with mortality among *S**. maltophilia* bacteremia patients include septic shock, APACHE II score ≥15 points, the indwelling of central venous catheters, mechanical ventilation, ICU admission, prolonged hospitalization, and receiving concurrent immunosuppressive agents or broad-spectrum antibiotics [19,20].

In our study, we identified that a high APACHE II score was also associated with higher rates of mortality, similar to previous literature [3,26,27,28]. Moreover, this study found that the appropriate empirical antibiotic(s) were a protective predictor to reduce mortality, based on the multivariate analysis results. However, some reports found that the inappropriate empirical antimicrobial treatment did not have a significant impact on mortality [12,18]. These discrepancies were probably due to the different antimicrobial susceptibility patterns. This study reported that TMP-SMX and levofloxacin remain highly active against isolates of *S**. maltophilia* (93% and 88%, respectively), whereas the susceptibility rates of TMP-SMX and fluoroquinolones have varied in some previous reports (81–85%, and 21–82%, respectively) [18,29].

With regard to the emergence of infections of *S**. maltophilia*, particularly in individuals who were admitted in ICUs, a previous report demonstrated the economic burdens among those patients with this infection including prolonged length of stay in hospital and hospital costs [30]. This current study did not only demonstrate only clinical benefits of appropriate antibiotics but also non-clinical benefits including length of hospital stay and hospital expenses. These findings suggest that the appropriate empirical antibiotic yielded an approximately 40% shorter length of hospital stay and an approximately 25% lower hospital cost. Even with the addition of appropriate empirical antimicrobial agents including fluoroquinolones and TMP-SMX into the commonly used regimens, the antimicrobial cost among those with appropriate empirical antimicrobial agents were still lower than those with inappropriate empirical antibiotics.

This study shows that *S**. maltophilia* mostly affects patients with a history of the prior use of antibiotics, especially carbapenem (96%), reflecting the widespread but potentially problematic use of carbapenem in hospitalized patients, which may lead to the emergence of *S**. maltophilia* infections due to its intrinsic resistance to carbapenem and the selective growth of non-susceptible isolates. Because *S**. maltophilia* infections continue to increase worldwide and therapeutic options are limited, antimicrobial stewardship is essential to reduce the emergence of drug-resistant organisms and hospital costs.

### Study Limitations

This study has some limitations. First, we enrolled patients from a single hospital, which could make it difficult to generalize our findings to other contexts. Second, the lack of data addressing the clinical judgment of physicians regarding antimicrobial therapy may have limited the scope of the analysis. Third, our study focused only on bloodstream infections and pneumonia, whereas other infection sites were not evaluated. Variations in the infection site might be associated with the outcome of *S**. maltophilia* septicemia. Lastly, the design of this study was a retrospective cohort analysis, which may have contributed to information and selection biases. A well-designed prospective study should be conducted to establish the impact of appropriate antimicrobial therapy on mortality outcomes more precisely.

## 5. Conclusions

Bacteremia due to *S**. maltophilia* causes relatively high mortality and high resource utilization. This infection should of concern among patients with several medical comorbidities, comparatively severe clinical manifestations, with previous admission to intensive care units and previous exposures to broad-spectrum antimicrobial agents, particularly carbapenems. Appropriate empirical antimicrobial treatment yields a significant benefit on clinical outcomes including 14-day, 30-day and in-hospital mortality. Additionally, the patients who received appropriate empirical antimicrobial treatment had a substantially shorter length of hospital stay and lower hospital costs. According to antimicrobial susceptibility testing in this current study, the appropriate empirical antimicrobials agents include TMP-SMX and fluoroquinolone.

## Figures and Tables

**Figure 1 jcm-11-03085-f001:**
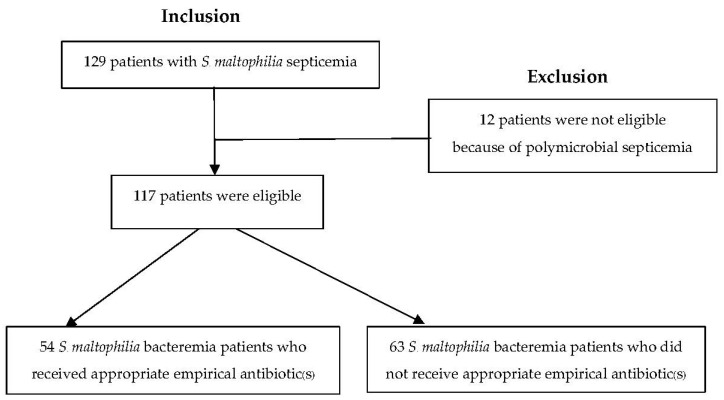
Study enrollment flow chart.

**Figure 2 jcm-11-03085-f002:**
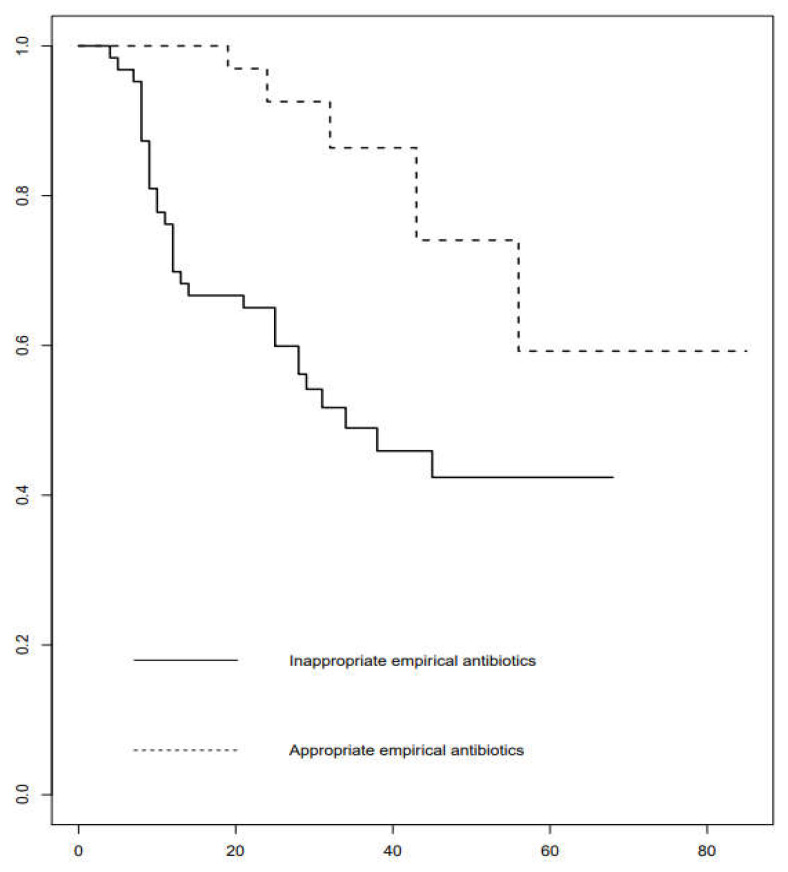
Kaplan–Meier survival curves of *S**. maltophilia* bacteremia patients who received and did not receive appropriate empirical antibiotic(s).

**Table 1 jcm-11-03085-t001:** Characteristics of *S*. *maltophilia* bacteremia patients and comparison of features between patients receiving and not receiving appropriate empirical antibiotics.

Parameter	*S*. *maltophilia* Bacteremia Patients (n = 117)	*S*. *maltophilia* Bacteremia Patients Receiving Appropriate Empirical Antibiotics (n = 54)	*S*. *maltophilia* Bacteremia Patients Not Receiving Appropriate Empirical Antibiotics (n = 63)	*p*-Value ^A^
Demographics				
Age, median (IQR)	45 (40, 51)	45 (36, 51)	45 (41, 51)	0.974
Male sex	77 (70)	40 (74)	37 (59)	0.121
Comorbidities	105 (90)	51 (94)	54 (86)	0.213
Immunocompromised status	6 (5)	3 (6)	3 (5)	0.997
Obesity	83 (70)	36 (67)	47 (75)	0.460
Previous exposure to antibiotics				
Carbapenem	112 (96)	50 (93)	62 (98)	0.158
Cephalosporin	85 (73)	41 (76)	44 (69)	0.462
Fluoroquinolone	72 (62)	32 (59)	40 (63)	0.639
β-lactam/β-lactamase inhibitor	69 (59)	36 (67)	33 (52)	0.119
Aminoglycoside	40 (34)	18 (33)	22 (35)	0.857
Clinical characteristics				
Initial ICU admission	54 (46)	32 (59)	22 (35)	0.014
APACHE II score, median (IQR)	19 (15, 23)	20 (13, 22)	16 (14, 18)	0.021
Pneumonia	55 (47)	26 (48)	29 (46)	0.966
Invasive medical devices				
Mechanical ventilator	95 (81)	48 (89)	47 (74)	0.054
Intra-vascular device	73 (62)	39 (72)	34 (54)	0.044
Urinary catheterization	101 (86)	52 (96)	49 (78)	0.008
Treatment Empirical treatment including				
Carbapenem(s)	110 (94)	50 (93)	60 (94)	0.551
Colistin	40 (34)	30 (56)	10 (15)	<0.001
TMP-SMX	30 (25)	30 (56)	0 (0)	<0.001
Fluoroquinolone(s)	31 (26)	28 (52)	3 (5)	<0.001
Duration of empirical treatment	3 (3, 4)	3 (3, 4)	3 (3, 4)	0.996

^A^ Comparison between *S**. maltophilia* bacteremia patients receiving and not receiving appropriate empirical antibiotics. Boldface entries indicate values that reached the significance level set at 0.05.

**Table 2 jcm-11-03085-t002:** Antibiotic-resistant profiles of 117 *S*. *maltophilia* isolates were obtained from the patients with bacteremia.

Antibiotics	No. of Resistant Isolates (%)
Chloramphenicol	21 (18)
Colistin	41 (35)
Levofloxacin	14 (12)
TMP-SMX	8 (7)
Ceftazidime	76 (65)

**Table 3 jcm-11-03085-t003:** Comparison of outcomes between patients with *S**. maltophilia* bacteremia who did or did not receive appropriate empirical antibiotic(s).

Outcomes	*S*. *maltophilia* Bacteremia Patients Who Received Appropriate Empirical Antibiotic(s) (n = 54)	*S*. *maltophilia* Bacteremia Patients Who Did Not Receive Appropriate Empirical Antibiotic(s) (n = 63)	*p*-Value ^A^
Clinical outcomes			
Mortality			
14-day	4 (7)	21 (33)	0.001
30-day	4 (7)	30 (48)	<0.001
In-hospital	5 (9)	32 (51)	<0.001
Non-clinical outcomes			
Length of hospital stay (days) [median (IQR)]	21 (15, 31)	34 (29, 50)	<0.001
Cost (baht) [median(IQR)]			
Total hospital	142,463 (125,008–212,389)	185,663 (159,223–200,685)	0.002
Antimicrobial	30,854 (26,447–33,321)	31,854 (29,606–34,014)	0.031
Non-antimicrobial	118,007 (97,245–175,004)	146,897 (131,488–190,884)	<0.001

^A^ Comparison between patients with *S**. maltophilia* bacteremia who did and did not receive appropriate empirical antibiotic(s). Boldface entries indicate values that reached the significance level set at 0.05.

**Table 4 jcm-11-03085-t004:** Factors associated with 30-day mortality in 117 *S**. maltophilia* bacteremia patients.

Variables	Values		Crude OR (95% CI)	Adjusted OR (95% CI)	*p*-Value ^A^
	Survivors (n = 83)	Non-Survivors (n = 34)			
Age (years) [median (IQR)]	45 (42, 67)	47 (36, 73)	1.01 (0.99, 1.03)	1.01 (0.96, 1.02)	0.386
Male sex	57 (69)	20 (59)	0.65 (0.28, 1.49)	0.53 (0.25, 2.15)	0.567
Underlying disease(s)	76 (92)	29 (85)	0.53 (0.16, 1.81)	0.3 (0.10, 1.05)	0.615
Immunocompromised status	5 (6)	1 (3)	0.45 (0.05, 4.12)	0.41 (0.08, 3.71)	0.738
Obesity	58 (70)	25 (74)	1.19 (0.49, 2.04)	1.06 (0.35, 2.00)	0.905
APACHE II score [median (IQR)]	16 (13, 21)	18 (15, 21)	1.05 (1.01, 1.15)	1.19 (1.04, 1.35)	0.012
Pneumonia	40 (48)	15 (44)	0.84 (0.53, 2.63)	0.72 (0.50, 2.46)	0.546
Initial intensive care unit admission	38 (46)	16 (47)	1.05 (0.47, 2.32)	1.03 (0.25, 1.54)	0.131
Appropriate empirical antibiotic(s)	50 (60)	4 (12)	0.09 (0.03, 0.27)	0.04 (0.01, 0.16)	<0.001

^A^ Comparison between survivors and non-survivors. Boldface entries indicate values that reached the significance level set at 0.05.

## Data Availability

The data presented in this study are available on request from the corresponding author. The data are not publicly available due to the privacy and ethical reasons.

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
