# Peer review of "Clinical Characteristics, Outcomes, and Risk Factors for Mortality in Patients with Stenotrophomonas maltophilia Bacteremia"

_jcm, 2022, doi:10.3390/jcm11113085_

Round 1

Author Response

Dear Editor,

We are very thankful for the valuable comments and suggestions from reviewer and editorial team. We made every effort to formulate appropriate responses to the comments and suggestions. The responses are as follows:

Comments from the Editor and Reviewers:

Reviewer 1.

Critical remarks and recommendations:

  1. L. 76-78 “The Kirby-Bauer disk diffusion method was used for antimicrobial evaluation. The results were interpreted according to the Clinical and Laboratory Standards Institute (CLSI)”.
  • According to the CLSI criteria, the antimicrobial susceptibility of S. maltophilia to trimethoprim-sulfamethoxazole, levofloxacin and minocycline can only be interpreted by the diffusion disk method.
  • S. maltophilia possesses intrinsic resistance to multiple antimicrobial agents including carbapenems (imipenem and meropenem) and aminoglycosides (amikacin and gentamicin) therefore, these antibiotics should be excluded from Table 2.
  • Regardless of the in vitro results obtained, S. maltophilia isolates should be interpreted and reported as resistant to aminoglycosides.
  • Antimicrobial susceptibility testing to ceftazidime and chloramphenicol is possible only by methods for determining the minimum inhibitory concentrations (MICs) according to the CLSI criteria used by the authors.
  • In summary, I ask the authors to find a way to correctly present the results presented in the text and Table 2.

Response: We apologized for the error. We have added the sentences in part of Bacterial Identification and Antimicrobial Susceptibility as follows: “The Kirby-Bauer disk diffusion method was used for antimicrobial evaluation of TMP-SMX, levofloxacin, and minocycline, while the Broth microdilution method was performed when testing ceftazidime and chloramphenicol. The results were interpreted according to the Clinical and Laboratory Standards Institute (CLSI) [21-22]. Regarding the polymyxin susceptibility test, there are no currently established polymyxin breakpoints for S. maltophilia, but Minimal Inhibitory Concentration (MIC) of ≥4 µg/ml is often considered resistance [23].”

We also removed meropenem, imipenem, amikin, gentamicin, piperacillin/tazobactam, and ciprofloxacin from the text Line 190-196 and Table 2 as follows:

“The results indicated that tigecycline, TMP-SMX, levofloxacin, doxycycline, and chloramphenicol showed the highest susceptibility rate against S. maltophilia, with susceptibility values of 94%, 93%, 88%, 87%, and 82% respectively, compared with the 65% and 35% non-susceptible rate of ceftazidime and colistin.”

Table 2. Antibiotic-resistant profiles of 117 isolates of S. maltophilia were obtained from the patients with bacteremia.

Antibiotics

No. of resistant isolates (%)

Doxycycline

15 (13)

Chloramphenicol

21 (18)

Tigecycline

7(6)

Colistin

41 (35)

Levofloxacin

14 (12)

TMP-SMX

8 (7)

Ceftazidime

                       76 (65)

  1. The discussion is too short and shallow. This section needs to be rewritten!

Response: We apologized for the error. We have revised the part of discussion as follows: “Several characteristics of S. maltophilia bacteremia patients were similar to previous reports. Several reports showed the infection due to S. maltophilia commonly occurred among the critically ill patients who were admitted in ICUs while only less than half (46%) of the patients in this research were primarily admitted in ICUs [3-8, 12]. It can be explained with the constrained ICU capacity in our setting. The patients in this study had relatively severe illness with the median APACHE score of 19 and most were subjected to mechanical ventilator (82%), intravascular devices (62%) and urinary catheterization (86%), respectively. Thus the acquisition of this organisms frequently occurred among mechanically ventilated patients [12]. Most of the patients in this study (90%) had several medical morbidities. Similarly, risk factors of this infection including the multiple debilitating illnesses were described in previous report. Additionally, previous exposure of several broad-spectrum antibiotics were previously demonstrated as the risk factor for infection due to S. maltophilia. This study shows that S. maltophilia mostly affects patients with prior use of antibiotics, especially carbapenem (96%), reflecting the use of carbapenem in patients who are hospitalized may lead to the emergence of nosocomial infections caused by S. maltophilia as its intrinsic resistance to carbapenem and the selective growth of non-susceptible isolates. 

Antimicrobial susceptibility test in this study showed unfavorable resistance to commonly-used antimicrobial agents for nosocomial infection including ceftazidime, and colistin. These findings were similar to previous published data [10-11]. The resistance seemed relatively low among levofloxacin (12%), TMP-SMX (7%), tigecycline (6%) and doxycycline (13%). However, tigecycline had too low serum level to suggestive to treat bacteremia [25]. In our setting, there was only oral form of doxycycline available which is not suitable for treating the bacteremic patients with critical illness. [26]. Then the appropriate choices in our setting included TMP-SMX and fluoroquinolones. It is interesting that the resistance of colistin is relatively high of 35%. This finding is synchronized with the reports in several regions with the rising of colistin-resistant S. maltophilia [10-11, 14]. Unlike to in vitro study of Enterobacteriaceae, the data on combination therapy of S. maltophilia remains unclear [27].

Previous reports have demonstrated that risk factors associated with mortality among       

  1. S. maltophilia bacteremia patients include septic shock, APACHE II score ≥15 points, the

      indwelling of central venous catheters, mechanical ventilation, ICU admission, prolonged  

      hospitalization, and receiving concurrent immunosuppressive agents or broad-spectrum

      antibiotics [19,20]. In our study, we identified that a high APACHE II score was also associated with higher rates of mortality, similar to previous literature [3, 28-30]. Moreover, this study found that the appropriate empirical antibiotic(s) was a protective predictor to reduce mortality, based on the multivariate analysis results. However, some reports found that the inappropriate empirical antimicrobial treatment did not have a significant impact on mortality [12,18]. These discrepancies were probably due to the different antimicrobial susceptibility patterns. This study reported that TMP-SMX, and levofloxacin remain highly active against isolates of S. maltophilia (93% and 88%, respectively), whereas the susceptibility rates of TMP-SMX and fluoroquinolones have varied in some previous reports (81-85%, and 21-82%, respectively) [18,31]. 

According to the emergence infections of S. maltophilia, particularly in individuals who were admitted in ICUs. Previous report demonstrated the economic burdens among those patients with this infection including prolonged length of stay in hospital and hospital costs [32]. This current study did not only demonstrate only clinical benefits of appropriate empirical antibiotics but also non-clinical benefits including length of hospital stay among the survivals and hospital expenses. The finding suggested the appropriate empirical antibiotic yielded approximately 40% shorter length of hospital stay and approximately 25% lower hospital cost. Even the addition of appropriate empirical antimicrobial agents including fluoroquinolones and TMP-SMX into the commonly used regimens, the antimicrobial cost among those with appropriate empirical antimicrobial agents were still lower than those with inappropriate empirical antibiotics. 

This study shows that S. maltophilia mostly affects patients with prior use of antibiotics, especially carbapenem (96%), reflecting the widespread but potentially problematic use of carbapenem in hospitalized patients, which may lead to the emergence of S. maltophilia infections due to its intrinsic resistance to carbapenem and the selective growth of non-susceptible isolates. Because S. maltophilia infections continue to increase worldwide and therapeutic options are limited, antimicrobial stewardship is essential to reduce the emergence of drug-resistant organisms and hospital costs.”

  1. Only 8% (2/24) of the literature sources included in the reference list have been published in the last 5 years. I highly recommend finding newer references and improving the list.

Response: We apologized for the error. We have added newer references now.

  1. The content of Figure 1 should be presented in 1-2 sentences in the text or its quality should be improved.

            Response: We apologized for the error. We had added the sentence in part of the text

            of Figure 1 as follows: “Figure 1 depicts patient enrollment in this study. With a total   

            117 patients with S. maltophilia septicemia, 54 (46.2%) patients received appropriate

            empirical antibiotics.”

Sincerely yours,

Sarunyou Chusri, M.D., Ph.D.

Reviewer 2 Report

  1. Line 36: bacterium, not bacteria
  2. Line 78: Add a reference for CLSI
  3. Line 88: Any patients receiving DMARD’s (MTX, MMF, AZA etc.) should also be considered immunocompromised
  4. Line 103: Continuous variables should be tested with a test like D’Agostino & Pearson’s in order to check for normality and if normally distributed, then t-test is indicated and data are usually presented as mean +/- SD. Otherwise, Mann-Whitney is the test that should be performed and data are usually presented as median (IQR)
  5. Figure 1: Something is wrong with the figure. The arrows are in the boxes
  6. Line 147: Add a space between the number and the percentage
  7. Line 150: What is [3,15-22]? They look like references. If yes, please remove them from the results section. If they depict something else, please make it clear
  8. The discussion section is somewhat small. It could be developed by including information regarding the magnitude of the problem of antimicrobial resistance, the need for antimicrobial stewardship in order to reduce infections by MDR, and the socio-economic magnitude of the problem

Author Response

Dear Editor,

We are very thankful for the valuable comments and suggestions from reviewer and editorial team. We made every effort to formulate appropriate responses to the comments and suggestions. The responses are as follows:

Comments from the Editor and Reviewers:

Reviewer 2

  Line 36: bacterium, not bacteria

    Response: We apologized for the error. We have changed the word “bacteria to bacterium”

  Line 78: Add a reference for CLSI

    Response: We apologized for the error. We have already added references for CLSI    

    in the part of Methods.

  Line 88: Any patients receiving DMARD’s (MTX, MMF, AZA etc.) should also be considered immunocompromised

    Response: We apologized for the error. We have revised the sentence in the part of Patients and Methods as follows: “Immunocompromised patients were defined as patients who had an absolute neutrophil count below 0.5 x 109/L for >2 weeks or receiving immunosuppressive therapy (chemotherapy within 6 weeks or corticosteroids at a dosage greater than 15 mg of prednisolone daily for >2 weeks or disease-modifying antirheumatic drugs within 4 weeks after the bacteremia onset)”

  Line 103: Continuous variables should be tested with a test like D’Agostino & Pearson’s in order to check for normality and if normally distributed, then t-test is indicated and data are usually presented as mean +/- SD. Otherwise, Mann-Whitney is the test that should be performed and data are usually presented as median (IQR)

Response: We apologized for the error. We have revised the sentences as follows: “Continuous variables were tested with the D’Agostino-Pearson test in order to check for normality and if normally distributed, then Student’s t-tests is indicated and data are usually presented as mean ± standard deviation (SD). Otherwise, Mann-Whitney is the test that should be performed and data are usually presented as median (IQR).”

  Figure 1: Something is wrong with the figure. The arrows are in the boxes

Response: We apologized for the error. We have corrected the Figure 1.

  Line 147: Add a space between the number and the percentage

Response: We apologized for the error. We have added a space between the number and percentage.

  Line 150: What is [3,15-22]? They look like references. If yes, please remove them from the results section. If they depict something else, please make it clear

Response: We apologized for the error. We have removed [3,15-22] from the result section.

  The discussion section is somewhat small. It could be developed by including information regarding the magnitude of the problem of antimicrobial resistance, the need for antimicrobial stewardship in order to reduce infections by MDR, and the socio-economic magnitude of the problem

Response: We apologized for the error. We have added the sentence in the part of discussion as follows: “Several characteristics of S. maltophilia bacteremia patients were similar to previous reports. Several reports showed the infection due to S. maltophilia commonly occurred among the critically ill patients who were admitted in ICUs while only less than half (46%) of the patients in this research were primarily admitted in ICUs [3-8, 12]. It can be explained with the constrained ICU capacity in our setting. The patients in this study had relatively severe illness with the median APACHE score of 19 and most were subjected to mechanical ventilator (82%), intravascular devices (62%) and urinary catheterization (86%), respectively. Thus the acquisition of this organisms frequently occurred among mechanically ventilated patients [12]. Most of the patients in this study (90%) had several medical morbidities. Similarly, risk factors of this infection including the multiple debilitating illnesses were described in previous report. Additionally, previous exposure of several broad-spectrum antibiotics were previously demonstrated as the risk factor for infection due to S. maltophilia. This study shows that S. maltophilia mostly affects patients with prior use of antibiotics, especially carbapenem (96%), reflecting the use of carbapenem in patients who are hospitalized may lead to the emergence of nosocomial infections caused by S. maltophilia as its intrinsic resistance to carbapenem and the selective growth of non-susceptible isolates. 

Antimicrobial susceptibility test in this study showed unfavorable resistance to commonly-used antimicrobial agents for nosocomial infection including ceftazidime, and colistin. These findings were similar to previous published data [10-11]. The resistance seemed relatively low among levofloxacin (12%), TMP-SMX (7%), tigecycline (6%) and doxycycline (13%). However, tigecycline had too low serum level to suggestive to treat bacteremia [25]. In our setting, there was only oral form of doxycycline available which is not suitable for treating the bacteremic patients with critical illness. [26]. Then the appropriate choices in our setting included TMP-SMX and fluoroquinolones. It is interesting that the resistance of colistin is relatively high of 35%. This finding is synchronized with the reports in several regions with the rising of colistin-resistant S. maltophilia [10-11, 14]. Unlike to in vitro study of Enterobacteriaceae, the data on combination therapy of S. maltophilia remains unclear [27].

Previous reports have demonstrated that risk factors associated with mortality among S. maltophilia bacteremia patients include septic shock, APACHE II score ≥15 points, the indwelling of central venous catheters, mechanical ventilation, ICU admission, prolonged hospitalization, and receiving concurrent immunosuppressive agents or broad-spectrum antibiotics [19,20]. In our study, we identified that a high APACHE II score was also associated with higher rates of mortality, similar to previous literature [3, 28-30]. Moreover, this study found that the appropriate empirical antibiotic(s) was a protective predictor to reduce mortality, based on the multivariate analysis results. However, some reports found that the inappropriate empirical antimicrobial treatment did not have a significant impact on mortality [12,18]. These discrepancies were probably due to the different antimicrobial susceptibility patterns. This study reported that TMP-SMX, and levofloxacin remain highly active against isolates of S. maltophilia (93% and 88%, respectively), whereas the susceptibility rates of TMP-SMX and fluoroquinolones have varied in some previous reports (81-85%, and 21-82%, respectively) [18,31]. 

According to the emergence infections of S. maltophilia, particularly in individuals who were admitted in ICUs. Previous report demonstrated the economic burdens among those patients with this infection including prolonged length of stay in hospital and hospital costs [32]. This current study did not only demonstrate only clinical benefits of appropriate empirical antibiotics but also non-clinical benefits including length of hospital stay among the survivals and hospital expenses. The finding suggested the appropriate empirical antibiotic yielded approximately 40% shorter length of hospital stay and approximately 25% lower hospital cost. Even the addition of appropriate empirical antimicrobial agents including fluoroquinolones and TMP-SMX into the commonly used regimens, the antimicrobial cost among those with appropriate empirical antimicrobial agents were still lower than those with inappropriate empirical antibiotics. 

This study shows that S. maltophilia mostly affects patients with prior use of antibiotics, especially carbapenem (96%), reflecting the widespread but potentially problematic use of carbapenem in hospitalized patients, which may lead to the emergence of S. maltophilia infections due to its intrinsic resistance to carbapenem and the selective growth of non-susceptible isolates. Because S. maltophilia infections continue to increase worldwide and therapeutic options are limited, antimicrobial stewardship is essential to reduce the emergence of drug-resistant organisms and hospital costs.”

Sincerely yours,

Sarunyou Chusri, M.D., Ph.D.

Reviewer 3 Report

Author has chosen very relevant topic and had nicely executed the study. I suggest the author to go through few comments highlighted in pdf attached. Only limitation ( as commented by author himself) is the retrospective design of the study. But based on the results of this study, author can continue this study in a prospective manner. 

Author Response

Dear Editor,

We are very thankful for the valuable comments and suggestions from reviewer and editorial team. We made every effort to formulate appropriate responses to the comments and suggestions. The responses are as follows:

Comments from the Editor and Reviewers:

Reviewer 3

Line 77: Which antibiotics were selected for disc diffusion? These are not matching with CLSI guidelines e.g. Aminoglycosides, shown in Table 2 are not recommended to test and report for Stenotrophomonas. Similarly, when it known that carbapenem are inherently resistant for Stenotrophomonas, what is the rationale for testing its susceptibility? 

Response: We apologized for the error. We have corrected the sentences in part of 2.2 Bacterial Identification and Antimicrobial Susceptibility as follows: “The Kirby-Bauer disk diffusion method was used for antimicrobial evaluation of TMP-SMX, levofloxacin, and minocycline, while the Broth microdilution method was performed when testing ceftazidime, chloramphenicol, and polymyxin. The results were interpreted according to the Clinical and Laboratory Standards Institute (CLSI) [21-22]. Regarding polymyxin susceptibility testing, there are no currently established polymyxin breakpoints for S. maltophilia, but Minimal Inhibitory Concentration (MIC) of ≥ 4 µg/ml is often considered resistance [23].”

We also removed meropenem, imipenem, amikin, gentamicin, piperacillin/tazobactam, and ciprofloxacin from the text and Table 2 as follows: “The results indicated that tigecycline, TMP-SMX, levofloxacin, doxycycline and chloramphenicol showed the highest susceptibility rate against S. maltophilia, with susceptibility values of 94%, 93%, 88%, 87%, and 82% respectively, compared with the 65% and 35% non-susceptible rate of ceftazidime and colistin. However, because the extensive distribution of tigecycline into the tissue, and blood levels are low, it is not recommended for the management of S. maltophilia bacteremia. Thus, TMP-SMX was adopted as the most effective antibiotic against S. maltophilia bacteremia, followed by fluoroquinolone. “

Table 2. Antibiotic-resistant profiles of 117 isolates of S. maltophilia were obtained from the patients with bacteremia.

Antibiotics

No. of resistant isolates (%)

Doxycycline

15 (13)

Chloramphenicol

21 (18)

Colistin

41 (35)

Levofloxacin

14 (12)

Trimethoprim-sulfamethoxazole

8 (7)

Ceftazidime

                       76 (65)

Line 92: Author should clearly define appropriateness of antibiotics for treatment. It will be better if the author can write what were the common empirical antibiotics given in this settings, for these infections.

Response: We apologized for the error. We have added the sentence in the part of Data collection as follows: “TMP-SMX, fluoroquinolones, and minocycline has been considered the treatment options for S. maltophilia infections [24].” And the empirical antibiotics given in this study were shown in Table 2. Carbapenem is the most common empirical antimicrobial choice which did not have activity against S. maltophilia as its intrinsic resistance. 

Line 140: There is overcrowding of data shown. Data not visible clearly.

Response: We apologized for the error. We have corrected the Figure 1.

Line 173: It should be “not receiving”

Response: We apologized for the error. We have corrected the word from “receiving to not receiving”

Sincerely yours,

Sarunyou Chusri, M.D.,Ph.D.

Round 2

Reviewer 1 Report

P. 3, L. 93-94 "The Kirby-Bauer disk diffusion method was used for antimicrobial evaluation. of TMP-SMX, levofloxacin, and minocycline, while the Broth microdilution method was performed when testing ceftazidime, and chloramphenicol. The results were interpreted according to the Clinical and Laboratory Standards Institute" - Where are the results regarding minocycline susceptibility?; You should exclude tigecycline and doxycycline (Table 2).

The conclusion paragraph is still unconvincing and too short. Please modify it!

Author Response

Dear Editor,

We are very thankful for the valuable comments and suggestions from reviewer and editorial team. We made every effort to formulate appropriate responses to the comments and suggestions. The responses are as follows:

Reviewer 1

Comments and Suggestions for Authors (Round 2)

  1. 3, L. 93-94"The Kirby-Bauer disk diffusion method was used for antimicrobial evaluation. of TMP-SMX, levofloxacin, and minocycline, while the Broth microdilution method was performed when testing ceftazidime, and chloramphenicol. The results were interpreted according to the Clinical and Laboratory Standards Institute" - Where are the results regarding minocycline susceptibility; You should exclude tigecycline and doxycycline (Table 2).

Response: We apologized for the error. We have removed “minocycline, tigecycline, and doxycycline” from text and table as follows: “The Kirby-Bauer disk diffusion method was used for antimicrobial evaluation of TMP-SMX and levofloxacin, while the Broth microdilution method was performed when testing ceftazidime, and chloramphenicol. The results were interpreted according to the Clinical and Laboratory Standards Institute (CLSI) [21-22]. Regarding polymyxin susceptibility testing, there are no currently established polymyxin breakpoints for S. maltophilia, but Minimal Inhibitory Concentration (MIC) of ≥4 µg/ml is often considered resistance [23].”

Antimicrobial susceptibility testing of the S. maltophilia isolated from the 117 patients is shown in Table 2. The results indicated that TMP-SMX, levofloxacin, and chloramphenicol showed the highest susceptibility rate against S. maltophilia, with susceptibility values of 93%, 88%, and 82% respectively, compared with the 65% and 35% non-susceptible rate of ceftazidime and colistin. Thus, TMP-SMX was adopted as the most effective antibiotic against S. maltophilia bacteremia, followed by fluoroquinolone. 

Table 2. Antibiotic-resistant profiles of 117 S. maltophilia isolates were obtained from the patients with bacteremia.

Antibiotics

No. of resistant isolates (%)

Chloramphenicol

21 (18)

Colistin

41 (35)

Levofloxacin

14 (12)

TMP-SMX

8 (7)

Ceftazidime

76 (65)

The conclusion paragraph is still unconvincing and too short. Please modify it!

Response: We apologized for the error. We have modified the conclusion paragraph as follows:

“Bacteremia due to S. maltophilia causes relatively high mortality and high resource utilization. This infection should be concerned among patients with several medical comorbidities, comparatively severe clinical manifestation, admission in intensive care units and previous exposure to broad-spectrum antimicrobial agents, particularly carbapenems. Appropriate empirical antimicrobials treatment yields significant benefit on clinical outcomes including 14-day, 30-day and in-hospital mortality.  Additionally, the patients who received appropriate empirical antimicrobial treatment had substantially shorter length of hospital stay and lower hospital costs. According to antimicrobial susceptibility testing in this current study, the appropriate empirical antimicrobials agents include TMP-SMX and fluoroquinolone.”

Sincerely yours,

Sarunyou Chusri, M.D., Ph. D.

Reviewer 2 Report

The manuscript has been improved now.

Author Response

Dear Editor,

We are very thankful for the valuable comments and suggestions from reviewer and editorial team. We made every effort to formulate appropriate responses to the comments and suggestions. The responses are as follows:

Reviewer 2

Thank you for your comment. We have sent the manuscript for English language editing.

Sincerely yours,

Sarunyou Chusri, M.D., Ph. D.
